# Management of Strategic Risks for the Sustainability of SMEs in the Manufacturing Sector in Antioquia

Andrea Jiménez *, Yennifer Arrieta, Maria Antonia Nuñez * and Eduart Villanueva 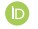

School of Management, Universidad EAFIT, Medellin 050022, Colombia; arrietajennifer12@gmail.com (Y.A.); evillanu@eafit.edu.co (E.V.)
* Correspondence: jimenezcastrillonandrea@gmail.com (A.J.); mnunezpa@eafit.edu.co (M.A.N.)

**Abstract:** Strategic risk management impacts organizations' competitive advantage; it is an opportunity for growth, anticipation, and sustainability. Small and medium-sized enterprises (SMEs) are significantly exposed to these risks and are a fundamental part of the current business fabric. This research aims to analyze the management of strategic risks for the sustainability of SMEs in the manufacturing sector in Antioquia. A qualitative methodology was used for the development, and ten semi-structured interviews were conducted with managers of the selected SMEs. The results show that strategic risk management contributes to the sustainability of SMEs across economic, social, and environmental pillars. The strategic risks most frequently mentioned as priorities for sustainability include long-term risks associated with strategy formulation and definition, as well as the value proposition. Additionally, the risk of human talent management, which is present in all analyzed organizations, was highlighted. Based on the research, interested organizations can be recommended to implement processes and practices associated with strategic risk management for sustainability. The research presents a significant novelty by specifically addressing the management of strategic risks for the sustainability of SMEs in the manufacturing sector. Unlike previous research, our study focuses on the specific risk management practices of SMEs in this geographical context, adding an important regional and sectoral dimension to the field of study.

**Keywords:** risk management; strategic risks; SMEs; sustainability; manufacturing sector; qualitative methodology

## 1. Introduction

The current scenario in economic, environmental, and social terms shows the constant alterations of the environment and the significance of these changes in the results, which increasingly affect financial well-being, maturity, market confidence, and reputation [1]. Latin America, for example, is facing environmental changes, inequality, and increasing violence, which generates constant uncertainty for people and, therefore, for companies [2]; these face challenges, threats, and disruptive situations [3].

Under this scenario, risk management has emerged as a mechanism to counteract uncertainty and has enabled companies to achieve objectives [4] and maximize value [5]. Specifically, strategic risk management is fundamental, given that when these risks are not managed in good time, some of their residual parts can accumulate over time, generating much more essential risks that become out of control in the future [6]. Strategic risks refer to the possibility that the formulation or strategic actions do not generate the expected results or, on the contrary, generate an opportunity [7]. Management of these impacts the organization's competitive advantage through growth opportunities, the ability to prepare in advance, and the ability to respond to economic, political, and social crises [8,9].

Given this context, organizations must implement strategic risk management to improve performance. Sustainable companies are those that have superior value—in economic, social, and environmental terms—to the market. They seek to reduce the negative

impacts and improve the positive effects of the actions carried out in the organization; thus optimizing natural, economic and human resources [10,11]. In this sense, it is essential to focus these analyses on small and medium-sized enterprises (SMEs) since they are the most significant part of the business fabric and are fundamental for growth, innovation, employment, social inclusion, and social sustainability [12,13].

Following on from the above, the research question is: how do SMEs in the manufacturing sector in Antioquia manage strategic risks for sustainability? The research aims to analyze the management of strategic risks for the sustainability of SMEs in the manufacturing sector in Antioquia; to respond to this, ten interviews were conducted with risk managers or leaders of SMEs. Initially, the strategy of each organization is understood. From there, strategic risks are derived, and then how the organization manages strategic risks from the identification, analysis evaluation, and treatment actions, in addition to the challenges associated with risk management for SME organizations; finally, the relationship between strategic risk management and sustainability is discussed.

The first section contains a literature review of strategic risks, their management, and their relationship with sustainability; this is followed by a description of the methodological aspects of the research, the results, and their analysis concerning the theory; and, finally, the conclusions are presented.

## 2. Literature Review

In this literature review, the conceptual approach to strategic risks is initially presented; this is followed by risk management, from which strategic risk management is broken down; and, finally, the relationship between strategic risk management and sustainability is discussed.

### 2.1. Strategic Risks

This concept is derived from the union of risk and strategy. Risks are understood as the effect of uncertainty on objectives; this definition invites us to study risks from the perspective of the effect of deviation, which can be positive or negative and recognizes not only the losses that may materialize but also the opportunities; from this perspective, the spectrum of risk management is increased [14]. Strategy is a coordinated and integrated set of five elements: a winning aspiration, where to play, how to win, core competencies, and administrative systems [15].

From the business perspective, there are different types of risks, and these can be classified according to their nature, the generating agent, consequence, organizational level, or treatment. For this research, we will delve into the organizational risks related to the level at which they occur. These are operational risks and strategic risks. Operational risk refers to the possibility of losses in executing the company's processes and functions due to failures of internal processes, people, systems, or external events [16,17]. Strategic risk is related to the possibility that the formulation or strategic actions do not generate the results expected by the organization or that, on the contrary, create an opportunity [7].

Strategic risks have been studied over the last few years by different actors, such as academics, consulting firms, and international organizations [18]; some associate them with the probability of deviation in the achievement of strategic objectives, others connect them with the uncertainty of the strategy, and others with the consequences of the choices made by the decision-maker of the organizations; finally, some define them as risks that threaten the survival of the organization [19]. Most studies or research developed around them have taken a positivist epistemological approach, presenting applied analyses and experimental processes with observation within a context [20].

To deepen the conceptualization of strategic risks, some authors define them as external events that can destroy the growth of the organization and its value to shareholders [21]. On the other hand, some authors specify them as those internal and external situations that can prevent the achievement of the strategy that is responsible for generating value to shareholders and stakeholders [22]. Others conceive them as uncertainties and untapped

opportunities integrated into the strategic intent and its respective implementation [7]. The latter definition is established as the basis for the present research.

Strategic risks can occur at two stages: strategy formulation and strategy implementation. During the strategy formulation process, top management, organizational teams, and other external stakeholders are involved, in addition to an external context that involves trends, the national and local environment, and the industry; this is why both internal and external risks can be generated in this process [19]. For some authors, risks arise in the strategy's implementation and affect the company's valuation [19]; they are related to the losses that occur from inadequate definitions in the strategy and are exhibited as the business objectives are developed [19]. For other authors, these can not only generate a decrease in revenue or losses but are risks that affect the profit and loss statement, so they can also generate a decrease in losses or an increase in profits [19]; this risk is not only understood with a focus on danger but also on opportunity [23]. Formulation and implementation risks can materialize in parallel; it is essential to constantly ask questions that allow reorientation of the strategy or taking corrective actions in the face of organizational changes [18].

The following are some of the strategic risks that may arise in the formulation of the strategy. Technological risk is understood as the possibility that new developments and technological advances may impact the environment, the population, and the economy and have repercussions on the company's business [19]. Another risk identified at this stage is that of error/mistake in the formulation of the strategy, which is the possibility of error in the definition of the organizational strategy [19]. Another risk identified at this stage is that of error in the design of the value proposition, which is defined as the possibility that the value proposition does not solve or satisfy the needs or requirements of customers [24]. In addition, the risk of change in customer preferences is identified; this refers to abrupt changes in the needs, tastes and priorities of customers that may affect the organization in the exercise of its activity [25].

In order to continue with the description of strategic risks, we will now address some of the risks that arise in the implementation of the strategy. Regulatory risk refers to the possibility that the organization is not prepared for regulatory changes applicable to the company, which may require a change in the decision scenarios [26]. The risk of misalignment in the deployment of the strategy is the possibility that the strategy formulated is not correctly implemented [27]. Liquidity risk is the possibility of not having the necessary cash flow to meet the company's financial obligations [28]. Another risk is that of ungovernability, which is the possibility that the government does not have sufficient capacity to solve economic, political, geopolitical, and/or environmental problems, which has direct effects on the organization [29,30]. The risk of competition is the possibility of strengthening or expansion of current competitors, lack of knowledge of competitors and the existence of unfair competitors that can change the company's value creation bases or leave it outside the market [31]. The risk of supplier dependence is the possibility of depending on one or several suppliers, which limits or even eliminates the organization's bargaining power; it can increase prices and decrease quality; and it can also increase the possibility that the supplier will become a competitor of the company [32].

Finally, due to its causes, intermediate risks may arise, i.e., from the formulation and deployment of the strategy; an example of this is the risk of human talent management, which is understood as the possibility of not having competent human resources aligned for the formulation and execution of organizational guidelines [26].

### 2.2. Strategic Risk Management

Risk management is a systematic and continuous process that includes resources, procedures, and practices [33]. Given the relevance of risk management in the achievement of objectives, an appropriate model is required; one of which is proposed by ISO 31000 [14], which is an international standard for risk management that consists of 3 sections. The first of them is the principles that must be integrated into the structure and objectives of the organization, which are aimed at creating and protecting value. The second section is

the frame of reference leveraged on leadership and commitment, for which it is necessary to integrate, design, implement, assess, and improve risk management. Finally, the risk management process must systematically implement policies, procedures, and practices for communication and consultation activities; in addition, the context must be established, the identification, evaluation, formulation of treatment actions, monitoring, review, registration, and reporting of the risk must be carried out [14]. These stages aim to identify actions that minimize the likelihood of adverse effects [34].

The positive impact of organizations with a risk management system on Tobin's Q indicator, which determines the value of the company's assets concerning the market, has been demonstrated [5]. It has also been proven that companies can achieve their objectives and maximize their value through risk management. Therefore, it should be integrated into the strategy [4]. The adoption of risk management with an integrated approach has become increasingly relevant [35], replacing siloed work [36]; this trend is called ERM (Enterprise Risk Management), which is a process led by senior management and boards of directors that is applied in the design and deployment of a strategy. An ERM system is a valuable tool that managers should implement to manage corporate reputation and maintain high performance in corporate social responsibility [37]. It is designed to identify potential events that may affect the organization and manage the risk to an acceptable level to ensure the company's objectives are met [34].

Another model for risk management was developed by COSO, the Committee of Sponsoring Organizations of the Tradeway Commission. It encompasses enterprise risk management and applies to both strategy and to the entire organization [38]. This framework sets out five pillars and 20 principles; the first pillar is governance and culture, it aims to establish the mission, vision, core values, oversight responsibilities, ethical values, desired behaviors, and understanding of risk in the organization; associated with this pillar, one of the success factors in a risk management system is top management leadership [36]. The next pillar is strategy and objectives, where the risk appetite is established, and it is emphasized that enterprise risk management must be aligned with the strategy and goals. The third pillar is performance; in the development of the plan, risks that may affect compliance with the plan are identified, and these risks must be prioritized by their severity and must be within the appetite. The next pillar is a review; it is necessary to monitor performance and evaluate how well the components of the enterprise risk system are working over time, in light of substantial changes. The last pillar corresponds to information, communication, and reporting; it is a continuous process to obtain and share necessary information, from internal and external sources, flowing in all directions and throughout the organization [34].

COSO, following its definition of risk as the possibility of some events occurring and affecting the achievement of strategic and business objectives, opened discussions about managing strategic risks. There is a call for systemic management of strategic risks to create growth opportunities, prepare in advance, and respond to crises [39]. Management is an opportunity for organizations; it was identified that the effect on strategic objectives is mediated by knowledge and early attention to strategic risk [40], and positive consequences can be generated for the growth and sustainability of the organization [19].

The relevance of strategy in organizations makes strategic risks a management focus for top management; these have a high potential impact on profits and, therefore, on the value perceived by the shareholder. The negative materialization of these risks indicates wrong strategic decisions or poor response to the requirements of the environment [41]. Likewise, proactive management of these risks can generate positive or negative deviations; it is considered that strategic risk management is the organization's response to the uncertainties and opportunities identified in the construction and development of the strategy through the understanding of the gap between the risk taken and the one that is wished for to make timely decisions [7]. In addition, senior management increasingly evidences expectations from shareholders, regulators, raters, and other stakeholders to internalize and manage strategic risks [38].

Strategic risk management has gained prominence in the organizational context to enable sustainability [42]; according to a recent analysis, financial stability is strengthened in companies with risk management tools [43]. Although some aspects hinder the management of strategic risks—such as uncertainty due to the high incidence of external risks, possible lack of transparency of decision makers, lack of experience, and complex development of treatment actions [44,45]—the adequate management of strategic risks can represent significant benefits, such as the reduction of production costs, regulatory compliance, better relationships with stakeholders, and competitive advantages for the organization [45].

*2.3. Sustainability and Strategic Risk Management*

Sustainability in the organizational sphere has been analyzed from different perspectives. It has been categorized as a complex and multidimensional concept [46] which represents organizations that are seeking to generate value, both from their strategic definition and their operational practices, to achieve results in terms of profitability and optimal management of resources, with the involvement of all stakeholders [47]. In developing their economic activity, most organizations present concerns related to economic, social, and environmental aspects, which are summarized in the great challenge of organizational sustainability [10].

Organizational sustainability seeks to reduce the negative impacts and improve the positive effects of the actions carried out in the organization. This is achieved by optimizing natural, economic, and human resources, and identifying and managing the risks associated with sustainability [11]. This becomes evident when there are results superior to the market in the pillars of sustainability (economic, social, and environmental) through coherent management of resources with all stakeholders in the conditions of the corresponding environment [48].

Companies working towards sustainability can improve their resilience, which is the capacity of an organization to adapt and recover to develop new skills and resources to the extent that they manage their risks and opportunities [49]. By simplifying the theories of uncertainty and chaos in strategy, it is possible to identify that by accessing and acting on information related to the pillars of sustainability, organizations improve the management of risks and opportunities [49,50]. Thus, a bidirectional relationship is found between risk management and organizational sustainability; the former leverages the achievement of positive impacts in economic, social, and environmental terms; and the latter relates to the extent to which the organization acts on its pillars, thereby improving risk management.

Some authors propose a bidirectional relationship between strategic risks and sustainability and, from there, establish the term sustainability risk, defining it as threats and opportunities that the organization has for its effort toward sustainable strategic development [45]. In addition, it is proposed that sustainability risks are usually highly variable in the long term; with a low probability of occurrence and high severity due to the connection with the strategy; consequently, their identification should be made about stakeholders based on internal and external elements and trends [45]. This definition of sustainability risk is highly connected to the definition of strategic risks, with similar attributes, and it is currently a priority for organizations to involve sustainability in the definition and materialization of the strategy [51].

Proactively managing strategic risks impacts the organization's competitive advantage in the dimensions or pillars of sustainability: economic, social, and environmental. First, it has been identified that, given the magnitude and impact of these risks, companies need to maintain reserves of economic capital to mitigate them and ensure their ability to adapt to changes and competitiveness [45]. These terms, as mentioned, are characteristic of a sustainable organization. Many studies have approached the quantification of the impact of strategic risks in economic terms by defining the capital required or the associated losses to support the risks; it has been identified that these risks usually have a high impact on the economic and financial capacity of organizations [8,9]. It has also been established that the

management of strategic risks is a critical factor in organizations, which affects profitability, financial strength, and commercial success in the medium and long term [35].

Regarding the social and environmental pillars, given the growth of these phenomena, organizations have increasingly considered them in the management of strategic risks, both from their description and their impact, i.e., there are environmental or social risks that are related to the formulation and execution of the strategy, as well as those that, if they materialize, may cause damage to the environment or society in general. It has been shown that there is currently a positive relationship between environmental and social performance, and the financial performance of organizations. Including these two pillars has become necessary for organizations to be competitive and remain in force over time [46].

## 3. Methodology

This research aims to analyze the management of strategic risks for the sustainability of SMEs in the manufacturing sector in Antioquia. Qualitative research focuses on understanding and detailing phenomena from the subjective perspective of the participants about the context [52], which is the input for the research development. The scope is descriptive, given that it expects to accurately show the dimensions around the object of analysis [52]. The collection of information was carried out using semi-structured interviews through a protocol that was tested in advance, which can be found in the final appendix of the article (see Appendix A). This method has gained much participation in generating knowledge; it allows one to approach how the subjects understand the phenomena and has validity due to the possibility of deepening or clarifying the data collection process [53].

The protocol was constructed by the literature review, initiated with the research objectives, in addition to critical definitions of the fundamental terms strategic risks, strategic risk management, and sustainability; this allowed the interviewees to have clarity about the framework of the analysis. The academic purposes of the interview and the questions based on the analysis categories were also considered. The interviews were conducted with the executives or managers of the companies; the data were obtained in the participants' natural environments and are mainly perceptions based on individual experiences or those of the corresponding organizations.

The categories of analysis used seek to highlight the central theme of the research; they were extracted from the literature review and are: strategic risks; strategic risk management (identification, analysis and evaluation, treatment actions, challenges of strategic risk management); and the relationship of sustainability with strategic risk management. After the interviews were conducted, they were transcribed and coded to analyze the information.

The study implemented careful measures to safeguard the rights, safety, and welfare of research participants, ensuring compliance with essential ethical standards. This included prioritizing the protection of human subjects from potential harm. Respect was given to principles such as informed consent and confidentiality. Data processing practices and protective measures were employed to uphold the rights and well-being of participants.

*Selection of the Participating Companies*

SMEs are essential for growth, innovation, employment, social inclusion, and social sustainability [12]. This global reality is fundamental for Latin America and Colombia, where the production of goods and provision of services is mainly represented by SMEs [13]. In Colombia, according to DANE, SMEs generate 78% of the jobs in the country and represent 99% of the companies in the country [54]. The survival outlook for organizations is complex; according to a study conducted in 2019, in the first year of life, 3 out of 4 companies survive (77.6%); and in the fifth year, 1 out of 2 companies remain in business (53.4%) [55].

Large companies have an advantage over SMEs due to their access to better-quality resources [56]. It was identified that implementation of sustainability-related practices is hindered in SMEs due to a lack of resources [57–59]. SMEs have different sources of disturbances that generate deviations in the organization's results and jeopardize its ability to survive [60,61]; the lack of financial management and management planning does not

mobilize the development of this type of organization. [61]. Given that SMEs are more exposed to changes in their environment [62,63], adopting sustainable practices should be incremental [64].

The transition to sustainability involves significant threats and opportunities for the industry in product development [45]. The manufacturing industry faces high pressures due to market freedom, changes in customer preferences, new legislation, and technological changes [65]. Their leading daily challenges are related to the resources to carry out their operations. However, planning and future perspective are necessary due to the demands of the environment, since they have a decisive role in the transition to a sustainable society [66]. For manufacturing companies to move towards sustainability, they require intelligent risk management [46], which involves more significant efforts in quality systems, adaptability, and innovation in their processes [46].

Notably, the manufacturing industry has a fundamental role in the development of Antioquia/Colombia. From an analysis between 2019 and 2021 of SMEs in Antioquia, it was identified that companies in the manufacturing sector have high participation in the business composition of the region and is where more Antioquian SMEs have been liquidated in the last three years, while their financial indicators are in the most unfavorable position compared to the other sectors for the period analyzed [67]. The analysis consisted of two stages, one of which was reactive and the other from a different perspective. In the first stage, the identification of the sectors where more SMEs have been liquidated in the last three years was carried out. In the second stage, the relevance of each industry in several companies and the respective financial results were analyzed based on profitability, indebtedness, and liquidity indicators [67]. A normalization exercise identified that the manufacturing sector is where more Antioquian SMEs have been liquidated in the last three years (25% of the total number of liquidated SMEs), while its participation in several companies places it in the second position of relevance (20% of companies correspond to this sector). Its financial indicators are in the most unfavorable position compared to the other sectors for the period analyzed.

It was defined for the development of the research that the type of organizations to be analyzed are SMEs in the manufacturing sector, the participating companies are selected randomly, the invitation to participate is made, and ten companies are prioritized in order of response, belonging to the manufacturing sector and the most relevant economic activities in income, which are: textile and food. Additionally, it is validated that they meet the conditions of SMEs in terms of revenue between 23,563 UVT and 204,995 UVT [68] and that they have established the strategy in the organization.

Interviews were conducted until a sufficient diversity of ideas had been explored, and subsequent interviews or observations did not yield any new elements. This phenomenon is commonly referred to as reaching the saturation point. This resulted in a sample of ten manufacturing SMEs in the food, textile, and kerosene processing subsectors. Some of their features are described in Table 1 below and listed to ensure confidentiality.

**Table 1.** Characteristics of participating companies.

|  | Subsector | Economic Activity | Size | Location |
|---|---|---|---|---|
| E1 | Textile | Manufacture of garments, except fur apparel | Small | Medellín |
| E2 | Textile | Weaving of textile products | Small | Santuario |
| E3 | Textile | Manufacture of garments, except fur apparel | Small | Medellín |
| E4 | Textile | Manufacture of articles of apparel, except fur apparel | Small | Medellín |
| E5 | Textile | Manufacture of articles of textile materials, except garments | Small | San Antonio de Prado |
| E6 | Food | Manufacture of dairy products | Medium | Don Matías |
| E7 | Food | Manufacture of dairy products | Medium | Santa Rosa de Osos |
| E8 | Food | Processing of dairy products | Small | La Unión |
| E9 | Food | Meat and meat products processing and preserving | Medium | Medellín |
| E10 | Candle production | Manufacture and processing of essential chemical products | Small | Medellín |

Source: Own elaboration, 2023.

The information from the interviews allowed the purpose of this research to be fulfilled; that is, to collect information.

## 4. Results

The results are obtained from an analysis of how SMEs in the manufacturing sector manage strategic risks for sustainability; and from the collection of the perceptions of the leaders of the organizations on the relationship between strategic risk management and sustainability. The main strategic risks to which the organizations are exposed, the stages of the risk management process, and the connection of these with sustainability within the framework of the economic, social, and environmental pillars are identified. For the analysis of the interviews carried out, comparisons were made between the answers given by the interviewees in each category of study, which arise from the literature; in addition, a section corresponding to the strategy of the organizations is annexed, which is a fundamental element for the understanding of strategic risks. A summary of the main findings by category is presented below in Table 2:

**Table 2.** Consolidated results.

| Category | Results |
|---|---|
| Strategic risks | The risks most frequently mentioned by the interviewees:<br>• Human talent management risk.<br>• Regulatory risk.<br>• Competition risk.<br>• Risk of dependence on suppliers. |
| Strategic Risk Management | Identification<br>Companies with a risk management system use tools such as DOFA and PESTEL. One of them creates a risk matrix and updates it yearly with teams from the organization and external advisors.<br>The other companies do not have defined methodologies; however, managers or primary committees identify the challenges associated with the strategy. Some rely on other companies or trade organizations, employees, customer and supplier perceptions, or external consultants.<br><br>Analysis and Evaluation<br>Companies with a risk management system analyze impact and frequency.<br>The other companies prioritize risks based on the organization's current needs and market experience.<br><br>Treatment actions<br>Avoiding, preventing, protecting, and retaining treatment actions were found to be the most frequently mentioned strategic risks. |
| Strategic risk management challenges | The main challenges are:<br>• Limited financial resources.<br>• Lack of knowledge in risk management.<br>• Overestimation of the chances of success.<br>• Highly uncertain and frequently changing environment. |
| Sustainability and strategic risk management | Prioritized strategic risks to business sustainability:<br><br>• Inaccuracy in the formulation of the strategy.<br>• Mistake in the design of the value proposition.<br>• Mistake in the deployment of the strategy.<br>• Liquidity risk.<br>• Key resources risk.<br>• Human talent management risk.<br>• Risk of dependence on suppliers.<br>• Competition risk.<br><br>Benefits of risk management for sustainability:<br><br>• Anticipation of and preparedness for contingencies.<br>• Close management that has enabled the development of improvement opportunities.<br><br>Positive consequences of risk management for sustainability:<br><br>• Adequate management of financial resources (economic).<br>• Employability and stakeholder management strategies (social).<br>• Better disposition of resources (environmental). |

Source: Own elaboration, 2023.

To begin to analyze the results, it is essential to know each organization's strategy and thus understand its strategic risks. In general, the managers of the companies interviewed have identified elements of the strategy; E1, E5, E6, E7, E8, and E9 have a deliberate strategy, while the others are characterized by having an emerging strategy. It was found that the companies that have a planned strategy have definitions in the five elements that make up the strategy: purpose, market, value proposition, core competencies and capabilities, and administrative system; however, companies that have an emergent strategy do not have some of the elements above; E2, E3, E4, and E10 do not have core competencies or administrative systems.

### 4.1. Strategic Risks

Based on the strategy, we inquired about the organization's strategic risks; the following is a summary of the strategic dangers most frequently mentioned by the people interviewed in the Table 3.

**Table 3.** Strategic risks most frequently referenced by the companies analyzed.

| Strategic Risks | E1 | E2 | E3 | E4 | E5 | E6 | E7 | E8 | E9 | E10 |
|---|---|---|---|---|---|---|---|---|---|---|
| Human talent management risk | X | X | X | X | X | X | X | X | X | X |
| Regulatory risk | X | X | X | X | X | X | X | | | |
| Competition risk | | X | X | | | X | X | | X | X |
| Supplier dependence risk | | X | X | | | X | X | X | | |
| Technological risk | X | | | X | X | | X | | | |
| Ungovernability risk | X | | | | | X | X | | X | |
| Liquidity risk | | | | | X | | | X | | X |
| Risk of change in customer preferences | | X | X | X | | | | | | |
| Strategy formulation missteps/mistakes | X | | | X | | | | X | | |

Source: Own elaboration, 2023. The "X" represents the strategic risks mentioned by the people interviewed.

All the companies analyzed identified the risk of human talent management and when asked about the causes high staff turnover, competition, low competitive salaries, and informality were identified. Regarding the first, E6 points out that "for some time we have been experiencing a phenomenon of personnel turnover because we know that our location is very prone to many people migrating" (E6); E7 considers that "many of these young people prefer not to stay working, they do not like the work in the field" (E7); and also, regarding competition, the person in charge of risk management in E7 recognizes that "there are also other companies dedicated to the dairy industry that have more experience, that is more multinational, that offer any little extra thing and go with the competition". The other element that influences is the informality in the market, which limits the development of talent over time. E2 affirms that "there is a lot of informality, even the process of the fabric is formalized (...), but when the fabric comes in, the garment business is more informal" (E2); and E1 considers that "the garment labor force is decreasing because there is less and less of it. What can we call it? It is less attractive for young people to say: I am going to spend my whole life working for a minimum wage glued to a machine" (E1).

Another risk highlighted is regulatory risk, which was identified by E1, E2, E3, E4, E5, E6, and E7, concerning the causes, the changes in the sector's applicable regulations, tax requirements, and the capital requirement to be able to support the implementation of rules stand out. This risk makes it difficult for SMEs to be competitive with large organizations in the market; E4 points out: "To grow, there must be capital, a significant injection of capital and, behind that injection of capital, there is also everything that has to do with compliance with legal regulations required by our national government to be competitive in the market" (E4).

The risk of competition was identified by E2, E3, E6, E7, E9, and E10; the interviewees described the leading causes of risk for all subsectors as aggressive price competition and market atomization. In the textile market, there is a high turnover of products, which has meant that organizations are attentive to changes in competition and create highly similar

products; in the food subsector, disputes are created by human talent and customers. This is a relevant risk for SMEs, given that competition is not only among them but also with large companies with more significant financial muscle and aggressive strategies. E10 points out: "Competition worries us, but not because we are not able to face it, but because there is a competition that is economically very well supported (. . .) and they have sales policies that I would say are not healthy, because they almost give away the merchandise and have huge promotions, and we do not have much capital" (E10).

A risk present in several companies is the risk of supplier dependence, which the leaders of companies E2, E3, E6, and E7 mentioned. This risk is fundamental for the manufacturing sector since it is responsible for transforming products in the textile and food subsectors; the supplier generates a bottleneck because it limits the raw material, and prices vary significantly depending on availability. E5 (textile) mentions that: "getting raw material at this time is complicated, one goes to look for fabrics and it is difficult, it is practically necessary to pay cash, and when you find them they are at high prices" (E5); and E8 (food), highlights that "we had a complex situation with the issue of milk management, there was a shortage of milk, so buying it required very high prices (E8)".

Specifically for the textile subsector, there is a relevant risk: changes in customer preferences due to fashion trends, seasonality, weather conditions, and the influence of external markets. E3 says: "The risk that worries me the most is that fashion will end, that is, that there will come a point when people no longer wear dresses and the brand has been very much identified by the theme of dresses" (E3).

It is shown that SMEs in the manufacturing sector in Antioquia face a series of recurring strategic risks, including human talent management, regulatory complexity, aggressive market competition, supplier dependence, and changes in customer preferences. These risks pose significant challenges to the competitiveness and sustainability of businesses in the region, highlighting the need to address them proactively to ensure long-term success.

### 4.2. Strategic Risk Management

Strategic risk management must: establish the context; identify, evaluate, and formulate treatment actions; follow up, review, register, and report the risk. Of the 10 company leaders interviewed, it was found that E7 and E8 have a risk management system, i.e., they have documented and standardized the process; the others perform actions that are part of the system but are not formalized in the organization. The following are how the organizations carry out the identification, analysis, evaluation, and treatment actions for strategic risks.

### 4.2.1. Identification

When investigating how organizations identify or recognize strategic risks, it was found that companies with a risk management system (E7 and E8) do so through organizational tools. E7 uses the SWOT matrix (Strengths, Weaknesses, Opportunities, Threats), and E8 uses the PESTEL method (Political, Economic, Social, Technological, Environmental, Legal) for the identification of external risks, and there are periodic meetings with internal management to define strategic risks, Strengths, and Threats). E8 uses the PESTEL method (Political, Economic, Social, Technological, Environmental, Legal) to identify external risks; and there are periodic meetings with internal management to define internal risks based on the perceptions of these stakeholders, in addition to having a risk matrix. In both companies, the actors involved in this stage are the manager, their respective primary committee, and advisory teams, which accompany the strategic development of the organizations. This is how E7 describes it:

"There is a SWOT and a committee they meet monthly to look at the mission, vision, strategy, and operational plans. There, they talk about risks, define each leader's role and how they will address it" (E7).

E8 has been in charge of having a standardized process for risk identification, which covers strategic risks, operational risks, and financial risks. More than three years ago, the

first risk matrix was made with the management team, and it is updated every year together with the organization's teams. The internal control team that is part of the administrative management makes the first proposal, and then it is discussed with the other management (commercial, operations, human resources); they analyze whether the risks continue or have been modified and then they are validated and updated with the advisory team. Regarding this identification process, the person in charge emphasizes: "The idea is to continue growing, and now, let's say, one has more knowledge of the risks, and it is culturizing them" (E8).

The other companies do not have defined methodologies for the identification of strategic risks; however, E1, E6, and E9 have periodic spaces in the primary committees to expose the points that may affect the achievement of the objectives of the organization, whether external or internal. They also have relationships with other companies and trade institutions related to the sector; E6 additionally provides spaces with employees and suppliers, to collect information corresponding to the risks of the organization. In E2, E3, and E10, the identification of strategic risks is solely in the hands of the organization's manager. However, they complement their vision with external actors. In the case of E2, the perception of employees and suppliers is considered; in E3, external advisors are consulted; and in E10, the perception of clients and suppliers is considered.

This section examines how companies identify strategic risks, finding that those with risk management systems use tools such as the SWOT matrix and the PESTEL method, and hold regular meetings to define risks. However, some companies lack defined methodologies, relying on management perception and input from employees, suppliers, and external advisors. This highlights the importance of promoting a risk management culture for comprehensive risk identification and effective risk management.

4.2.2. Analysis and Evaluation

In the development of the interviews, questions about how the companies analyze and prioritize the risks were deepened. In the development of the methodology, companies with risk management systems (E7 and E8) evaluate the risks based on frequency and impact. The person in charge of risk management at E8 mentions regarding the process and evaluation criteria:

"I meet, in the company of the SIGA (Integrated Management and Self-Control System) coordinator, with each of the directorates and those directorates, in turn, with their work teams, and we define if the challenges that are continuing, what qualification is given to them because we have some measurement variables, which are: the probability of occurrence, the magnitude of the impact and the controls that are in place" (E8).

In the other companies, risks are not analyzed systematically but according to need, market knowledge, and, in some cases, as a result of consultancies received. In company E1, using quantitative and qualitative analyses carried out by the organization's leaders, actions are defined in the primary committees in addition to monitoring the indicators and, based on this, resources are allocated. In companies E3 and E5 these analyses are outsourced, and the managers of the organizations receive the respective evaluation inputs.

This section delves into how companies analyze and prioritize risks. Those with risk management systems evaluate risks based on frequency and impact, employing structured processes such as meetings with directorates and work teams to assess ongoing challenges, using variables such as probability of occurrence, impact magnitude, and existing controls. This systematic approach facilitates informed decision-making. In contrast, other companies analyze risks based on market knowledge, needs, and sometimes external consultations. This highlights the diverse approaches companies employ in risk assessment, which may impact the effectiveness of their risk management strategies and their ability to address business challenges.

### 4.2.3. Treatment Actions

The actions undertaken based on the identification of strategic risks were investigated. Below, the actions corresponding to the strategic risks most referenced by the companies classified in the types of treatment measures are presented in the Table 4.

**Table 4.** Strategic risk treatment actions.

| | Avoid | Prevent | Protect | Retain |
|---|---|---|---|---|
| Human talent management risk | | E3 and E5: Definition of roles and responsibilities of management and employees. E3, E5, and E8: Hiring of accounting, financial, and legal advisory services. E1 and E5: Training and education of employees and managers. E9: Talent retention strategy. | | |
| Regulatory risk | | E3, E5, and E8: Contracting of accounting, financial, and legal advisory services (constantly referencing the regulatory environment). | E3, E5, and E8: Contracting of accounting, financial, and legal advisory services (tax management and response to regulatory agencies). | |
| Supplier dependence risk | | E3. Payment agreements with suppliers. E7. Execution of supplier development program. E7. Supplier incentive program for quality improvements. | E7. Execution of supplier development program (price standardization). | |
| Technological risk | | E4 and E7. Design and implementation of the maintenance plan. | | E4. Financial reserve for maintenance contingencies and low production seasons. |
| Liquidity risk | | E2. Monitor and control the portfolio. | | E4. Financial reserve for maintenance contingencies and low production seasons. |
| Change in customer preferences risk | E2 and E4: Product line change | E3. Listen to customer preferences through social networks. | | |
| Risk of ungovernability | | | E6. Design contingency plans for environmental and public order emergencies. | |
| Strategy formulation error/mistake | | E1 and E6. Family protocol design. | | |

Source: Own elaboration, 2023.

When classifying the treatment actions, it was found that no actions were associated with transferring or pursuing strategic risks. Among the treatment actions that stand out is the one executed by E7, corresponding to the execution of the supplier development program. In this initiative, a system was created that works as circular economy, in which the company (E7) markets the inputs required for the development of the raw material, the supplier (producer of the raw material) buys them and then sells raw material

to the company. The strategy allows the supplier to have more accessible prices from the beginning, create a long-term relationship with the organization, and improve its socioeconomic conditions. The purpose of the initiative refers to:

"They have an area of livestock promotion; it is an area of negotiation and welfare, which is the one that is pending to try to maintain relations with these producers, to listen to their complaints, and to try to solve" (E7).

The organization ensures the availability of raw materials, builds supplier loyalty and reduces price volatility. It also contributes to shaping the social and environmental fabric of the region. This company also has an incentive program, which consists of providing feedback about the quality of inputs; this has generated motivation among suppliers and improved the quality of the final product.

Another of the treatment actions that stands out is the talent retention strategy, which according to the company E9, has generated promising results perceived in the decrease of the turnover indicator. The plan is composed of two elements: on the one hand, the company created policies for attraction, i.e., given an analysis that was carried out based on risk identification, they identified that the people who rotated the most met specific attributes; therefore, they established criteria for hiring based on age, nationality, knowledge, and skills; and on the other hand, they created new incentives and mechanisms for the development and retention of talent that is already in the organization. Regarding this point, E9 highlights:

"Our company has been generating talent retention strategies, such as additional benefits to what other companies similar to ours have, for example, two non-wage extralegal bonuses, discounts for company personnel in the purchase of products, other credit benefits directly with the company, direct contract and not through a temporary company, and also recreational benefits" (E9). The company constantly monitors risk and, as changes are perceived, makes adjustments to the actions implemented.

In the treatment category, it was observed that few actions were taken specifically to transfer or mitigate strategic risks, with most actions focusing on managing risks internally. Notably, one company implemented a supplier development program, fostering a circular economy model to ensure a stable supply chain and improve socioeconomic conditions for suppliers. This initiative not only secures raw material availability but also enhances supplier loyalty and reduces price volatility, contributing to regional social and environmental well-being. Additionally, another company implemented talent retention strategies, resulting in decreased turnover rates. These strategies include tailored hiring criteria and additional benefits to retain existing talent, demonstrating a proactive approach to mitigating human capital-related risks.

### 4.2.4. Risk Management Challenges

The most common reason among the leaders for which it is difficult for them to have a risk management system is financial resources. E1, E2, E4, and E10 agree with this argument; in particular, E1 highlights that their main limitation is "having the money to do it. It is not enough to think about tomorrow because we are still thinking about it today" (E1). E10 also states that "it is difficult because SMEs generally work more than anything else so that they can sustain themselves in the market because to move forward there must be financial muscle" (E10).

Another reason is the knowledge of risk management, which was started by E3, E4, E5, E7, and E8. SMEs, in some cases, do not have entrepreneurs in charge who are trained in risk management, the daily management of organizational concerns consumes the time that could be implemented to create such knowledge, or organizations overestimate the chances of success. Against this point E5 and E3 state:

"The limitation at this time of SMEs, for me, is the training of entrepreneurs" (E5). "We may not have so many worries, suddenly we are very positive and one never sees negative scenarios that something is going to happen, no, one tries to have everything

under control" (E3). E8 also mentions that risk management is limited because it must be at the forefront of what is happening in the market, which is highly changeable.

*4.3. Sustainability and Strategic Risk Management*

In the interviews, we inquired about the strategic risks that the organization has, but we were asked to prioritize these risks in terms of sustainability in the deepening of what are the priorities for the sustainability of the same. Most organizations have different perceptions depending on the context in which they operate. E1, E3, and E8 mentioned how their most critical risks for sustainability are: the risk of misguidance in strategy formulation, the risk of misguidance in the design of the value proposition, and the risk of misguidance in the deployment of the strategy. In all three companies, it is essential to know the long-term outlook of the organization. E2 considers that the most critical risk is liquidity risk; E4 considers the risk of crucial resources the most critical; E5, E6, and E9 identify the risk of human talent management as the most important; E7 highlights the risk of dependence on suppliers; and E10 the risk of competition.

Respondents were also asked about the benefits of strategic risk management in the sustainability of the organization. E1, E7, and E8 consider that the most relevant is the organization's ability to anticipate and be prepared for any contingency; as organizations create different scenarios and ready to respond to them, they will have more time and better capabilities at the time they materialize and, therefore, do not significantly sacrifice results, as will happen to those who are not prepared. E7 also considers that risk management has allowed close management that considers global and particular elements and has helped to identify opportunities for improvement.

In addition, the consequences generated by managing strategic risks on the organization's sustainability were consulted in economic, social, and environmental terms. In the first element, it was found that, according to E1, the management of strategic risks allows companies to anticipate and, therefore, have the liquidity and cash flow necessary to respond to contingencies that affect the development of their strategic objectives. E2 considers that the organization manages to be sustainable to the extent that there are financial resources, and that implies good portfolio management. E7 and E8 mention that adequate management of financial resources and achieving the break-even point are essential elements for sustainability. The following was stated by the person interviewed in E7:

"The management of financial resources, because sometimes there is waste, there is waste and returns that, if minimized and controlled, also impact natural resources, labor; because that is important for sustainability" (E7).

Some of the actions established consider the stakeholders. For example, E3, from the identification of the risk of dependence on suppliers, has made payment agreements that balance the financial welfare of the parties to keep them over time, reducing the variability of prices related to raw materials. E3, E5, and E8 have sought advice on financial issues to improve the development of management plans of the organization, and this outsourcing alternative has contributed to the management and identification of strategic risks, for the development of the organization.

It was also found that organizations develop actions related to social and environmental issues; however, not all are related to risk management, and some come from regulatory compliance or management aspirations. E1, from its strategic formulation, considers that the company's purpose is not financial; it is to generate jobs and build a country. E3 and E4 have sought to reduce the environmental impact of raw materials and design processes; E5, E6, E7, E8, and E9 have an environmental management system and are increasingly committed to compliance with environmental regulations and improving ecological conditions.

In general, risk management allows one to highlight elements that, if properly managed, lead to better organizational results. In the case of E7, mentioned above, the construction of the supplier development strategy arises from the identification of the risk of high

dependence on suppliers for the fulfillment of objectives; in this case, it is fundamental how the organization implements a strategy that contributes to the social development of the region, as well as to a better circulation of resources to achieve better performance in its results, i.e., it contemplates the three pillars of sustainability. Finally, the person interviewed in E8 explicitly states the importance of strategic risk management for sustainability:

"I would think that the priority risks for the sustainability of the organization are the strategic ones because the issue of structure, the issue of direction, lead to knowing where the organization is going, it is the strategic planning, hence if you do not know where you are going, it is difficult for things to go" (E8).

In the sustainability and strategic risks category, organizations were asked to prioritize strategic risks concerning sustainability. Findings revealed varied perceptions among companies, with risks ranging from misguidance in strategy formulation to liquidity risk, human talent management, and competition. The benefits of strategic risk management highlighted anticipation and preparedness for contingencies, enhancing organizational resilience. The consequences of managing risks were discussed in economic, social, and environmental terms, emphasizing the importance of financial resource management, stakeholder engagement, and environmental compliance. Actions taken by organizations reflected efforts to address risks and promote sustainability across economic, social, and environmental dimensions, underscoring the integral role of strategic risk management in organizational sustainability.

## 5. Discussion

The development of the research presents how SMEs in the manufacturing sector carry out strategic risk management and how this contributes to sustainability; for this purpose, the strategic risks that the companies have, the elements of risk management, the challenges of their development, and the relationship between risk management and sustainability were analyzed. When addressing the strategic risk management of these companies, similarities were found with the information consulted in the literature review of bibliographic references in the field. It was found that, in general, the interviewees understand the strategic risks to which they are exposed and recognize the importance of managing them for sustainability.

In the first place, when it comes to understanding these risks, many organizations consider that they are those that divert them from achieving their objectives, or that they are those of which they may be unaware or that do not allow the continuity of the business. Some of these companies have already faced bankruptcy or the closure of their operations and have associated the risks mentioned to these critical moments. The literature currently acknowledges a range of viewpoints regarding strategic risks, which may encompass both favorable and unfavorable outcomes. It not only acknowledges potential losses but also identifies opportunities. This broad perspective enhances the spectrum of risk management [14]. Some authors define them as the probability of deviation in the achievement of strategic objectives, also as the uncertainty of the strategy, or as the consequences of the choices of the decision-maker of the organizations; and, finally, there are those who establish them as the risks that threaten the survival of the organization [19].

Furthermore, the findings validate that strategic risks are defined as internal and external circumstances capable of hindering the realization of the strategy aimed at creating value for shareholders and stakeholders [22]. These identified risks are evident in both the formulation and implementation phases of the strategy [19]. For example, the risk of misjudgment/error in strategy formulation and, in the second case, the risk of dependence on suppliers. Likewise, according to the literature, that strategic risks depend on the organizational context [21] is evident that given the characteristics shared by the sector and the region where they are located. Many of these companies share risks that condition the fulfillment of their objectives. However, each has at least one particular risk for its context.

For the execution of strategic risk management, it was found that many of these companies do not have a formalized system. However, the literature states that risk man-

agement is a systematic and continuous process that includes resources, procedures, and practices [33]. However, they do implement some practices related to risk management. Within these, some methodologies were found for the identification of risks in the organization, follow-up of crucial figures and indicators, prioritization, and creation of corrective actions, which are combined, depending on the case. This coincides with the literature review, which establishes that, depending on the context, a technique or combination of these can be used [14]. Most companies have these activities of identification, evaluation, and development of treatment actions, headed by senior management or administrative or financial teams; and they are complemented by external advisory teams and other institutions or guilds. The literature establishes that the success of risk management is the leadership of senior management and its inclusion in organizations is increasingly required [38].

In the identification of strategic risks, two cases were presented of companies that use the DOFA and PESTEL tools, and the others make empirical analyses with work teams or supported by third parties. However, the literature recommends other tools that can be useful for the analysis of the external and internal context such as Porter's five forces analysis to analyze the industry, the Delphi method, scenario analysis, understanding the business model through the Osterwalder and Pigneur Canvas, and the analysis of resources and capabilities [19].

During the analysis and evaluation of strategic risks, it was identified that companies without risk management systems do not utilize techniques to analyze and assess risks, confirming what has been previously found in the literature and highlighting the importance of having comprehensive risk management systems [35]. However, the leaders, based on their knowledge, experience, or external support, determine the level of priority of the risks. The criteria for this decision are usually varied and are not formalized. On the other hand, companies that have strategic risk management systems analyze and evaluate their risks based on frequency and impact. This contrasts with the literature review, in which techniques are used to understand the probability or consequences of risks, such as risk registers, risk maps, probability-impact tables, risk matrices, failure rates, waiting models, and preliminary cause analysis [19].

Based on the strategic risks identified in each organization, treatment actions have been adopted around the principles of: avoid, prevent, protect, and retain. None of the organizations mentioned adopting actions related to transfer or pursue; the first refers to sharing the impact of the materialization with a third party and the second is to assume more significant risks in the search for new opportunities [14,34]. The companies consider that the establishment of treatment actions has allowed them to meet their organizational objectives and find exit mechanisms in complex situations, which is consistent with what is established in the literature, given that companies that implement risk management or have practices related to it state that they have positive impacts on the results of the organizations [5].

According to the literature, there are some challenges for risk management in organizations, such as the lack of experience or knowledge of the leaders in the field and uncertainty due to the incidence of external risks [44]. In unity with the literature, the people interviewed also point out knowledge and uncertainty among the elements that currently hinder strategic risk management. However, the main challenge mentioned by the interviewees is the difficulty in obtaining resources for development; this element is not mentioned in the referenced literature analysis, given that in this case the analysis is carried out on a business segment of small and medium-sized companies, which do not yet have the financial muscle to support this system.

Regarding the advantages of strategic risk management, as mentioned in the literature, the people interviewed recognize that it allows companies to anticipate and respond to crises [39]. Furthermore, the interviewees emphasized the importance of conducting effective management of these types of risks to support the achievement of companies' strategic objectives. This reinforces what is highlighted in the literature, indicating that

understanding and early attention to strategic risks benefit organizational objectives [40]. A relationship between risk management and sustainability is identified, and it is found that those risks prioritized from this approach are usually long-term, with a low possibility of occurrence and high severity due to the connection with the strategy [46].

The connection of strategic risk management with sustainability is marked in the three pillars: economic, social, and environmental. It is identified that the interviewees recognize from their practices that the organization's sustainability is related to all the pillars. However, the economic pillar predominates in most of them. It is found that companies that manage strategic risks manage to have better levels of liquidity and establishment of financial resources, i.e., financial stability [43]. Some actions were also identified that have been carried out by organizations that have an impact on environmental and social development, based on the management of strategic risks. Finally, it is highlighted that most companies refer to the need to consider the three pillars of sustainability to be competitive and remain in force over time, which is consistent with the literature [46].

## 6. Conclusions

The results of the research show the main conclusion that the management of strategic risks contributes to the sustainability of organizations in the economic, social, and environmental pillars. This represents a relevant finding when investigating the relationship between these two fields of study. It was found that the strategic risks most referred to by the managers of SMEs as priorities for sustainability are long-term risks, related to the formulation and definition of the strategy and the value proposition, as well as the risk of human talent management; the latter risk being present in all the organizations analyzed. It was also identified that some organizations do not have strategic risk management systems; however, this is not a limitation since they all have practices for identifying, analyzing, and implementing actions derived from strategic risks.

These analyses validate the need for organizations to strengthen their risk management practices from the top management to improve organizational results in different areas, especially in a scenario of high uncertainty currently affecting SMEs. It is also very relevant that academia and associations create and execute joint strategies for conceptual strengthening organizational leaders in managing strategic risks.

### 6.1. Practical Implications

The research has practical implications for SME organizations wishing to implement strategic risk management for sustainability. The study shows that companies with practices related to managing strategic risks can improve their results. These risks, being directly related to the fulfillment of objectives, are a focus of management in small and medium-sized organizations that currently have essential challenges in sustainability. In the development, some of the strategic risks most referenced by the leaders interviewed are presented. These can be similar to the companies that share characteristics of the analyzed sample; additionally, some of the practices and challenges that these companies have in the implementation of risk management are presented, which can be helpful in the creation of processes or implementation of good practices in the companies. Some of the strategies carried out by the SME organizations that connect with sustainability and that have helped them to have better performance in different areas are also presented, which becomes a fundamental element for the economic and social development of the region, given that SMEs currently represent the majority of the business fabric.

The development of the research also addresses one of the significant challenges that risk management currently faces in SMEs, and that is the lack of knowledge of the people who lead the organizations and their processes around risks and how to manage them. This research provides transversal elements that can be put into practice within the organizations.

*6.2. Theoretical Implications*

The research results show the importance of strategic risk management for organizations' sustainability within the economic, social, and environmental pillars. This allows uniting of two currents that are increasingly gaining strength in the academic field, given the relevance of sustainability and strategic risk management for the development of organizations. In addition, the analysis is carried out on a business segment that increasingly requires integration with academia for the economic and social development of the region.

*6.3. Limitations and Future Research*

The study acknowledges its limitations in focusing solely on companies from one emerging country. Future research should consider examining companies from various industries in developed countries to validate and expand upon these findings. Furthermore, the study is limited to the relationship between two fields of study such as strategic risk management and sustainability, and can be further expanded in other research to provide different complementary approaches to the topic. While our qualitative approach provides valuable data and insights, conducting further quantitative analysis would be beneficial to broaden the research. From the results obtained in this study, several opportunities for future research were also identified. One of them is to develop research with a quantitative approach to determine the impact of strategic risk management on the sustainability of the organization; another possible line is how to strengthen risk management from the social and environmental approach. In addition, to deepen in the stages of strategic risk management in SMEs, to promote the use of some tools; it is then proposed to work on the question of how it is possible to implement risk management with limited resources; this has significant importance because it was the main challenge highlighted by the companies, as well as the articulation of the practices that SMEs currently carry out that can have an impact on sustainability.

**Author Contributions:** Investigation, A.J., Y.A., M.A.N. and E.V.; Writing—review & editing, A.J., Y.A., M.A.N. and E.V. All authors have read and agreed to the published version of the manuscript.

**Funding:** The project did not receive funding. The APC payment is made by the authors.

**Institutional Review Board Statement:** Not applicable.

**Informed Consent Statement:** Informed consent was obtained from all subjects involved in the study.

**Data Availability Statement:** Data are contained within the article.

**Conflicts of Interest:** The authors declare no conflict of interest.

## Appendix A. Interview Protocol

1. What is your background in the organization?
2. What is the organization's strategy?
3. What are the organization's strategic risks?
4. How do you identify or recognize strategic risks?
5. Of the strategic risks mentioned above, which ones do you consider to be a priority for sustainability? Mention the reasons.
6. What actions have been taken based on the identification of strategic risks?
7. What are the organization's current capabilities and resources to manage its strategic risks?
8. What are the positive and negative consequences of strategic risk management on the sustainability of the organization? In economic, social, or environmental terms.
9. What are the organization's strategic risk management challenges?
10. What benefits do you consider that strategic risk management would have on the sustainability of the organization?

The information provided is confidential, for academic purposes and the results of the research will be sent for your knowledge.

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
