# Peer review of "Management of Strategic Risks for the Sustainability of SMEs in the Manufacturing Sector in Antioquia"

_sustainability, doi:10.3390/su16052094_

Round 1

Reviewer 1 Report

Comments and Suggestions for Authors

Dear authors, congratulations on the study presented.

In my opinion, the topic is relevant, the methodology is appropriate and the work they present has scientific rigor that justifies its publication.

The literature review is pertinent, adequate and recent.

In the methodology, the authors present the reasons why the study focuses on the SME of Antioquia, as well as the strategy they followed to obtain the data.

The presentation and discussion of results is in line with the initially identified objectives.

The suggestions I make are minor and only aim for specific improvements.

The introduction starts with an example. This should not be the way to guide the reader towards a problem that is intended to be studied. First, they must present arguments and then reinforce them with examples.

At the end, they should also include the limitations of the study.

Good luck for future work.

Author Response

Dear reviewer 1,

We appreciate your time to review our paper and all your comments that allowed us to improve our text. Below we relate each of your comments to the response in blue.

Comments and Suggestions for Authors

  1. Dear authors, congratulations on the study presented.
  2. In my opinion, the topic is relevant, the methodology is appropriate and the work they present has scientific rigor that justifies its publication.
  3. The literature review is pertinent, adequate and recent.
  4. In the methodology, the authors present the reasons why the study focuses on the SME of Antioquia, as well as the strategy they followed to obtain the data.
  5. The presentation and discussion of results is in line with the initially identified objectives.

The suggestions I make are minor and only aim for specific improvements.

  1. The introduction starts with an example. This should not be the way to guide the reader towards a problem that is intended to be studied. First, they must present arguments and then reinforce them with examples.

R/ The change is accepted and the introduction is modified

  1. At the end, they should also include the limitations of the study.

R/ The suggestion is accepted and the limitations of the study are presented. This can be seen in point 6.3

In the attached file you can see the paper with the modifications

Thank you very much.

Kind regards,

Authors

Reviewer 2 Report

Comments and Suggestions for Authors

This paper examines “Management of strategic risks for the sustainability of SMEs in the manufacturing sector in Antioquia”. Below I give my comments.

1- The inner part of the abstract is indeed not so clear and requires careful reworking.

2- More meaningful keyword list should be prepared

3- Novelty of the work need to be explained more detailed.

4- Please compare critically the provided results with the results or fit parameters provided in the literature.

5- This manuscript lacks systematic theoretical analysis and discussion. It is suggested to give more detail in theoretical.

6- The methodology is not clear. Please, explicate better the steps you adopted. Also, the aims and the gap in literature are not introduced. Key results need to be compared with the literature with more discussion.

7- Numerical Results and discussion section requires significant revision to ensure that it presents clear interpretation of the results.

8- Results and discussion section should give more comparison.

Comments on the Quality of English Language

Minor editing of English language required

Author Response

Dear reviewer 2,

We appreciate your time to review our paper and all your comments that allowed us to improve our text. Below we relate each of your comments to the response in blue.

Comments and Suggestions for Authors

This paper examines “Management of strategic risks for the sustainability of SMEs in the manufacturing sector in Antioquia”. Below I give my comments.

1- The inner part of the abstract is indeed not so clear and requires careful reworking.

R/ The inner part of the abstract was reviewed and proofreading was done to provide greater clarity.

2- More meaningful keyword list should be prepared

R/ The suggestion is accepted and new keywords were included to encompass the purpose and meaning of the study.

3- Novelty of the work need to be explained more detailed.

R/ Details explaining the novelty of the study were expanded in the last paragraph of the abstract and in the first paragraph of the conclusions.

4- Please compare critically the provided results with the results or fit parameters provided in the literature.

R/ The commentary was followed and comparisons of the results with the literature review were added. This can be seen in the discussion section.

5- This manuscript lacks systematic theoretical analysis and discussion. It is suggested to give more detail in theoretical.

R/ Added further analysis and discussion on the support of the literature. This can be seen in different sections of the text but especially in the discussion.

6- The methodology is not clear. Please, explicate better the steps you adopted. Also, the aims and the gap in literature are not introduced. Key results need to be compared with the literature with more discussion.

R/ The suggestion to explicitly include the objective of the research in the methodology was addressed. More details were given of the steps followed for the ethical use of the information. Furthermore, it was specified that the information was collected until the saturation point was reached. In the discussion section, comparisons between the key results found in the research and the existing literature on the topic were expanded upon.

7- Numerical Results and discussion section requires significant revision to ensure that it presents clear interpretation of the results.

R/ The suggestion was accepted and an interpretation of each of the categories was presented in the results section.

8- Results and discussion section should give more comparison.

R/ The comment is accepted and further comparison was made. This can be seen in the discussion section.

Comments on the Quality of English Language

Minor editing of English language required.

R/ Further revision of the English language was carried out.

In the attached file you can see the paper with the modifications

Thank you very much.

Kind regards,

Authors

Reviewer 3 Report

Comments and Suggestions for Authors

Dear authors, 

The research aims to propose the management of strategic risks for the sustainability of SMEs in the manufacturing sector in Antioquia, using a qualitative methodology composed of ten semi-structured interviews, which were conducted with managers of the selected SMEs. In my opinion, the article addresses an important research problem in a developing country where these studies are typically lacking. Moreover, the article is generally well estructured and argued. However, there are some points that I think could help improved the manuscript. 

1) The first time the authors use SME, it is advisable to describe what the acronym means. Most people know this refers to small and medioum sized entreprises. However, the article could be read by people outside the field of research. 

2) In line 54 is mentioned "conducted with risk managers or leaders of SMEs with the characteristics above". However, these characteristics are not explicitly stated in the text before this sentence, causing confusion. These characteristics are clearly stated in 3.1. section. Therefore, I would suggest removing this sentence or clearly stating that these characteristics will be explained later in the manuscript. 

3) From lines 60 to 68, the authors summarized the results of the paper. In my opinion, the introduction is not the place to summarize the results. This should be summarized in the abstract and perhaps in the conclusion to linked with the wide literature review. I would suggest removing this paragraph from the introduction. 

4) Methodologically, it would be advisable to include the semi-structured interview script as an annex.

5) Methodologically, It would be advisable to comment the saturation effect for qualitative research on this topic. The saturation effect indicate the point in qualitative data collection where no new or additional information, themes, or insights emerge. I would suggest including this point in the methodology and also comparing it with other similar studies.

6) In my opinion, the structured of table 2 is confusing. I would suggest reviewing it to make it clearer.    

7) In lines 517 to 520 the text is written in spanish. 

8) Normally, research that collects data from people should have the approval of an ethics committe, but it is not indicated in the methodology section. Please comment on the reason if it is applicable to this research. 

I hope my comments could help the authors improve their manuscript. 

Kind regards, 

Reviewer.

Author Response

Dear reviewer 3,

We appreciate your time to review our paper and all your comments that allowed us to improve our text. Below we relate each of your comments to the response in blue.

Comments and Suggestions for Authors

Dear authors,

The research aims to propose the management of strategic risks for the sustainability of SMEs in the manufacturing sector in Antioquia, using a qualitative methodology composed of ten semi-structured interviews, which were conducted with managers of the selected SMEs. In my opinion, the article addresses an important research problem in a developing country where these studies are typically lacking. Moreover, the article is generally well estructured and argued. However, there are some points that I think could help improved the manuscript.

1) The first time the authors use SME, it is advisable to describe what the acronym means. Most people know this refers to small and medioum sized entreprises. However, the article could be read by people outside the field of research.

R/ The suggestion is accepted and the meaning of the acronym is included. This can be seen in the abstract and introduction.

2) In line 54 is mentioned "conducted with risk managers or leaders of SMEs with the characteristics above". However, these characteristics are not explicitly stated in the text before this sentence, causing confusion. These characteristics are clearly stated in 3.1. section. Therefore, I would suggest removing this sentence or clearly stating that these characteristics will be explained later in the manuscript.

R/ The suggestion is accepted. This can be seen in the introduction.

3) From lines 60 to 68, the authors summarized the results of the paper. In my opinion, the introduction is not the place to summarize the results. This should be summarized in the abstract and perhaps in the conclusion to linked with the wide literature review. I would suggest removing this paragraph from the introduction.

R/ The suggestion is accepted. This paragraph is eliminated from the introduction and thus the conclusion is strengthened.

4) Methodologically, it would be advisable to include the semi-structured interview script as an annex.

R/ The interview script is included as an annex and related in the methodology.

5) Methodologically, It would be advisable to comment the saturation effect for qualitative research on this topic. The saturation effect indicate the point in qualitative data collection where no new or additional information, themes, or insights emerge. I would suggest including this point in the methodology and also comparing it with other similar studies.

R/ This information was added. This can be seen in section 3.1 of the methodology.

6) In my opinion, the structured of table 2 is confusing. I would suggest reviewing it to make it clearer. 

R/ The suggestion is accepted. The structure of table 2 was modified to provide more clarity. 

7) In lines 517 to 520 the text is written in spanish.

R/ The change was made.

8) Normally, research that collects data from people should have the approval of an ethics committe, but it is not indicated in the methodology section. Please comment on the reason if it is applicable to this research.

R/ This information was included in the methodology starting at line 306.

In the attached file you can see the paper with the modifications

Thank you very much.

Kind regards,

Authors.

Reviewer 4 Report

Comments and Suggestions for Authors

I congratulate the authors on their good scientific work. It is successful both theoretically and methodologically, providing new knowledge. The authors accurately formulated and then implemented the cognitive goal. They conducted good qualitative research, although the selection of surveyed entities may raise some debates. The results are presented in a very simple way, which may be an advantage of this work. The authors quoted many statements obtained from their interlocutors, which greatly enriches the presentation of the research results.

To sum up, I believe that the presented work is so mature and complete that it is worth appreciating it by publishing it in a wide-ranging journal. It is rare for me to evaluate such well-done scientific works, so I do not raise any comments and propose to publish it in its current form.

Author Response

Dear Reviewer 4,

We greatly appreciate your time to review our paper. Your comments make us very happy and motivate us to continue investigating.

Thank you very much.

Kind regards,

Authors.

Comments and Suggestions for Authors

I congratulate the authors on their good scientific work. It is successful both theoretically and methodologically, providing new knowledge. The authors accurately formulated and then implemented the cognitive goal. They conducted good qualitative research, although the selection of surveyed entities may raise some debates. The results are presented in a very simple way, which may be an advantage of this work. The authors quoted many statements obtained from their interlocutors, which greatly enriches the presentation of the research results.

To sum up, I believe that the presented work is so mature and complete that it is worth appreciating it by publishing it in a wide-ranging journal. It is rare for me to evaluate such well-done scientific works, so I do not raise any comments and propose to publish it in its current form.

Round 2

Reviewer 2 Report

Comments and Suggestions for Authors

According to me, the current status of the paper can be accepted for publication.

Reviewer 3 Report

Comments and Suggestions for Authors

Dear authors, 

In my opinion, the comments to the first revision have been addressed correctly. Congratulations! 

Kind regards, 

Reviewer.